# Effects of Saturation on Anger in a Low-Saturation Range: A Comparison of Background Colors in 12 Tones

**DOI:** 10.3390/bs15040551

**Published:** 2025-04-18

**Authors:** Akinori Shimodaira, Noriyuki Kida

**Affiliations:** 1Doctoral Program of Advanced Fibro-Science, Graduate School of Science and Technology, Kyoto Institute of Technology, 1 Hashikami-cho, Matsugasaki, Sakyo-ku, Kyoto 606-8585, Japan; 2Faculty of Arts and Sciences, Kyoto Institute of Technology, Kyoto 606-8585, Japan; kida@kit.ac.jp

**Keywords:** three attributes of color, hue, value, saturation, emotion, anger, tone, PCCS

## Abstract

This study used an online survey to investigate the effects of brightness in low-saturation color ranges on anger processing. Specifically, it explored how background hues—red, yellow-green, and blue-green—affect perceptions of illustrations of an angry red face. The experiment involved 36 color combinations classified into three hue groups and three saturation levels (high, medium, and low) based on the Practical Color Co-ordinate System. The results indicate that the influence of hue disappears in the low-saturation range. Across all the saturation levels, lower brightness intensified the perception of anger, with the anger elicited by darker colors similar in strength to that elicited from vivid red. These findings offer new insights into the role of color in emotional processing, particularly in relation to anger.

## 1. Introduction

In modern society, everyday communication is undergoing a major transformation in form. The rapid progress in information and communication technology has enabled individuals to be both senders and receivers of information. As we evolve from traditional text-based communication to complex messaging, which incorporates a wealth of visual elements, the use of illustrations and photographs is particularly noteworthy. Illustrations that mimic facial expressions, called “stamps” or “emojis”, have become widespread on messenger applications and social media. These illustrations can convey a wealth of information on their own, and by combining shapes and colors, senders can incorporate complex emotional expressions, which leads to these images sometimes supplementing or even completely replacing text. In this study, we focus on illustrations of faces among the various designs offered by stamps. Among these, within pre-installed pictogram sets on many devices, stamps that express anger are often red.

The strong influence of color on emotions is widely recognized. In a previous study ([8]), the emotional responses to five primary colors, five intermediate colors, and three achromatic colors were evaluated, and it was found that the primary colors evoked many positive emotional responses. Furthermore, [18] ([18]) investigated emotional responses to hue, saturation, and value, and reported that saturation and value, in particular, have a strong effect on emotions. In addition, it was shown that blue, blue-green, green, red-purple, purple, and violet-blue were the hues that induced the most pleasant feelings, while yellow-green, blue-green, and green evoked the most excitement, and violet-blue and yellow-red also caused high levels of excitement. [3] ([3]) placed a checkered pattern of red or green in the background of facial expressions with ambiguous emotional values, such as neutrality or surprise, and asked participants to judge whether the expression was positive or negative. The results showed that when the background color was red, the expressions were more likely to be perceived as negative than when a green background was used. The psychological effects of color are also substantial. For example, the emotional state of a man wearing red clothing is more likely to be classified as “anger” [19] ([19]). [6] ([6]) reported that facial illustrations using the typical red color for complexion increased feelings of anger in expressions other than anger, such as joy and sadness. [14] ([14]) investigated the effects of six background colors on anger in response to vivid color illustrations of angry faces in red and yellow-green, finding that anger was heightened when the background was red. Another study investigating the effects of red on detecting anger and facial expressions using color priming with red, green, and blue, which are similar to vivid colors, and gray, which is a neutral color, [22] ([22]), reported that using a red background enhanced anger detection.

However, we posit that research on the relationship between hue and emotion being mainly conducted on vivid colors is problematic. Colors have three attributes that can be used to classify them in detail: hue, value, and saturation. Hue indicates the type of color; value indicates the brightness of the color; and saturation indicates the vividness and purity of the color. The effects of value and saturation as well as hue need to be considered to fully understand the effects of color; however, as an example of a study that combined these elements with human emotions, [20] ([20]) investigated arousal to color through skin conductance and heart rate using 27 chromatic colors with three hues, three levels of value, and three levels of saturation, as well as 3 achromatic colors. Moreover, colors placed against a light background tend to look darker. This phenomenon is called the “contrast effect”, and it shows that the way an object appears can change depending on the colors and brightness of the surrounding area. The surrounding environment and background also affect perception. To gain a deeper understanding of the effects of color combinations and explore new methods of expression, the effects of color combinations on anger need to be clarified using various background colors. Therefore, in this study, we examined the effects of color on emotions from the perspectives of hue, value, and saturation.

In general, humans perceive vitality through color saturation. For example, if the saturation of a person’s complexion is high, they are perceived as having vitality, and their cheerfulness and emotional strength are conveyed. If saturation is low, they are perceived as having little vitality and often give the impression of poor health. [20] ([20]) confirmed that bright, saturated colors are associated with joy. Additionally, [10] ([10]) examined how the emotional valence (positive or negative) of a photographic image or negative affected responses to color and saturation and found that highly saturated colors increased pleasantness ratings for positive images and decreased pleasantness ratings for negative images. This suggests that saturation affects people’s emotional responses to images. Based on this finding, we predicted that highly saturated colors would express anger more clearly and increase anger in illustrations depicting anger. However, as how low-saturation colors affect anger remains unclear, this was the first question addressed in this study. Based on this question, we examined how colors in the low-saturation range affect anger.

Value is also important in the study of low-saturation ranges. In general, high-value colors not only give a bright and refreshing impression but also make a space feel larger and create an active atmosphere. However, low-value colors give a sense of calm and stability and create a sense of luxury and depth while simultaneously making a space feel smaller and creating a heavy or oppressive atmosphere. [13] ([13]) stated that colors with low saturation and low brightness are more likely to cause stress, and [20] ([20]) also indicated that dark colors with low saturation are associated with fear. These findings also suggest that the lower the value, the more likely it is to amplify negative emotions such as anger; however, the specific effects of this have not yet been fully elucidated. Furthermore, [4] ([4]) reported that the darkness of a color is linked to negative emotions, and [14] ([14]) suggested that low value may increase anger. However, in these studies, comparisons were made using vivid colors, but no numerical comparisons were made of the differences in values within the same hue. In this study, we used the three attributes of color (i.e., hue, value, and saturation) as numerical indices and set the second question to clarify how differences in value within the same hue in the low-saturation range affect anger. The purpose was to verify whether a low value in the low-saturation range increases negative emotions—especially anger.

To address these questions, we classified colors based on the tone categories of the Practical Color Co-ordinate System (PCCS, Japan Color Research Color System) and designed an experiment. Figure 1 shows the distribution of the value and saturation of the colors in this study. Specifically, vivid, bright, strong, and deep were classified as high-saturation colors (9s and 8s); light, soft, dull, and dark were classified as medium-saturation colors (5s); and pale, light grayish, grayish, and dark grayish were classified as low-saturation colors (2s). Furthermore, for the red complexion (v2) expressing anger, we adopted red (2), which most intensifies the anger emotion, and yellow-green (10), which weakens it the most, from among the eight background colors used in the previous study by [14] ([14]), and added blue-green (14), which is the complementary color of red. These background colors were selected from the 12 tones defined in the PCCS, and the impact of the differences in background color on the impression of the illustration expressing anger was compared using 36 color schemes. The HSV values for each background color are shown in Table 1. Figure 2 shows the 12 tones of the PCCS in terms of which saturation range they belong to, as well as the positions of red, blue-green, and yellow-green on the hue circle.

The concept of brightness in this study needs to be organized from two perspectives: luminance and perceptual brightness. While luminance is a quantitative indicator of physical brightness, perceptual brightness refers to the subjective sense of brightness perceived by the visual system and brain. This study uses the color-rendering system PCCS and brightness as an indicator of perceptual brightness. Human vision is sensitive to contrast effects, and the brightness perceived for the same color may change depending on the background conditions. For this reason, considering the effect of the contrast between complexion and background on perception is also necessary.

This method allows us to compare differences in brightness and saturation within the same hue. Verifying the effect of vivid red color on anger by considering the relationship between the three attributes of color (hue, brightness, and saturation) is also possible.

Previous research has shown that red evokes stronger anger than other hues in the high-saturation range ([14]; [22]). Conversely, research on the medium- and low-saturation ranges ([13]; [20]) is limited, and even fewer studies have examined the two-color color scheme effect of front and background colors. In particular, in the low-saturation range, colors approach gray; therefore, distinguishing between hues is expected to become more difficult, and verifying the effect of the value on the impression of anger in a low-saturation range will be easier. In this study, to examine the pure contrast effect of facial and background color, we decided that illustrations with black outlines, which were used in a previous study ([6]), could have a confounding effect owing to the intervention of a third color element; thus, we used illustrations without outlines.

## 2. Methods

### 2.1. Survey Participants and Procedures

Through Google Forms, a web survey was conducted with 129 participants living in Japan: 4 in their teens or younger, 8 in their 20s, 10 in their 30s, 76 in their 40s, 29 in their 50s, and 2 in their 60s or older. The sample included 26 male and 99 female respondents and 4 who did not respond. The time it took the respondents to answer the questions ranged from 5 to 10 min. The survey was conducted over a period of 7 days. In a preliminary pilot survey, we confirmed that the display did not affect responses for computers (MacOS 24-inch), tablets (iOS 11-inch, WindowsOS 10.1-inch), or smartphones (iOS 6.1-inch, AndroidOS 5.1-inch) (red face, ICC = 0.973), and the survey was conducted on the devices that the participants owned. The devices used for the responses were smartphones (*n* = 96), tablets (*n* = 8), and PCs (*n* = 25). This survey was conducted with people who have no color vision problems, but as the participants were recruited widely over the Internet, the presence or absence of color vision problems is based on the participants’ own perceptions. The display format for the diagram when the survey was conducted is as shown in Figure 3.

### 2.2. Colors Used and Survey Items

In this study, we selected the vivid tone of red (v2) as the facial color for the angry facial expression. Additionally, we selected red (2), which was the background color that most enhanced the angry emotion, and yellow-green (10), which was the background color that most weakened the angry emotion, from the eight background colors used in the previous study by [14] ([14]). In addition to these two colors, we also selected three hues: blue-green (14), which is the opposite color of red. During the question survey, six images were displayed for each hue, and 12 tones from the PCCS (vivid, bright, strong, deep, light, soft, dull, dark, pale, light grayish, grayish, and dark grayish) were arranged randomly in a way that did not suggest the order of the tones (Figure 3). The images were then displayed so that they could be compared simultaneously, and participants were asked to rate how much “anger” they felt from each image. Ratings were made on a 5-point scale: 1 = *I don’t feel anger*, 2 = *I feel it a little*, 3 = *I feel it*, 4 = *I feel it strongly*, and 5 = *I feel it very strongly*.

### 2.3. Ethical Considerations

In accordance with the approval of the Kyoto Institute of Technology’s Research Ethics Committee for Human Subjects (approval number 2023-52) and the Ethical Code of the Japanese Society of Applied Psychology, we indicated, on the start page of the survey, that the respondents would not be forced to answer any questions they did not want to answer, that their responses would not be used for any purpose other than the research, and that the data would be anonymous and no individual identified. We obtained responses only from those who provided their consent.

## 3. Results

The mean and standard deviation for each combination of hue and tone are shown in Table 2, broken down by gender. The results of a three-way analysis of variance with hue, tone, and gender as factors show that interactions involving gender (hue × tone × gender: *F*_44, 2772_ = 0.669, *p* = 0.954, *η*^2^ = 0.011; tone × gender: *F*_22, 2772_ = 0.713, *p* = 0.830, *η*^2^ = 0.011; hue × gender: *F*_4, 2772_ = 0.388, *p* = 0.817, *η*^2^ = 0.006, and the main effect of gender (*F*_2, 126_ = 1.801, *p* = 0.169, *η*^2^ = 0.028) were not significant. Next, a significant result was obtained for the interaction between hue and tone (*F*_22, 2772_ = 5.368, *p* < 0.001, *η*^2^ = 0.041). We, therefore, analyzed the simple main effects using the Bonferroni method (Table 3, Figure 4), and first examined the differences between the tones for each hue. The results showed that for the vivid tone, red was the highest, and significantly higher than the other two colors (blue-green, *p* = 0.027; yellow-green, *p* < 0.001). For the bright tone, red was the highest, but there was no significant difference with blue-green, and it was only significantly higher than yellow-green (*p* = 0.037). In the strong tone, red was also the highest, but there was no significant difference from blue-green, and it was significantly higher than yellow-green (*p* = 0.004). In the deep tone, red was the highest, and it was significantly higher than the other two colors (blue-green, *p* = 0.001; yellow-green, *p* < 0.001). In the light tone, red was the highest, and there was a significant difference compared to yellow-green (*p* = 0.028), but there was no significant difference compared to blue-green. In the soft tone, red was the highest, but there was no significant difference compared to the other two colors (blue-green and yellow-green) (*p* > 0.050). In the dark tone, red was the highest, but there was no significant difference between it and the other two colors (blue-green, *p* < 0.001; yellow-green, *p* < 0.001). In the pale tone, red was the highest, but there was no significant difference between it and the other two colors (blue-green, *p* > 0.050; yellow-green, *p* > 0.050). In light grayish tone, red was the highest, but there was no significant difference with the other two colors (blue-green and yellow-green) (*p* > 0.050). In the grayish tone, red was the highest, and there was no significant difference with yellow-green, but it was significantly higher than blue-green (*p* = 0.012). In the dark grayish tone, red was the highest, but there was no significant difference between the other two colors (blue-green and yellow-green) (*p* > 0.050). Next, we examined the differences between hues by tone. In the background color blue-green, dark grayish was the highest, significantly higher than all the other tones (vivid, *p* < 0.001; bright, *p* < 0.001; strong, *p* < 0.001; deep, *p* < 0.001; light, *p* < 0.001; soft, *p* < 0.001; dull, *p* = 0.022; dark, *p* = 0.019; pale, *p* < 0.001; light grayish, *p* < 0.001; grayish, *p* < 0.001). Other combinations that showed significant differences included between vivid and light (*p* = 0.005), soft (*p* < 0.001), between bright and dull (*p* = 0.005), dark (*p* = 0.008), strong and pale (*p* = 0.045), deep and light (*p* = 0.041), soft (*p* = 0.002), pale (*p* = 0.002), light grayish (*p* = 0.009), grayish (*p* = 0.029), light and dull (*p* < 0.001), dark (*p* = 0.002), soft and dull (*p* < 0.001), dark (*p* < 0.001), dull and pale (*p* < 0.001), light grayish (*p* < 0.001), grayish (*p* = 0.001), dark and pale (*p* < 0.001), light grayish (*p* < 0.001), and grayish (*p* < 0.001). In the red background color, dark grayish had the highest value. Although there were no significant differences between vivid, deep, and dark, it was significantly higher than the other tones (bright, *p* < 0.001; strong, *p* = 0.036; light, *p* < 0.001; soft, *p* < 0.001; dull, *p* < 0.001; pale, *p* < 0.001; light grayish, *p* < 0.001; grayish, *p* < 0.001). Other combinations that showed significant differences were vivid and light (*p* = 0.027), soft (*p* = 0.004), dull (*p* < 0.001), pale (*p* < 0.001), and light grayish (*p* < 0.001); bright and deep (*p* = 0.004), dark (*p* = 0.044), pale (*p* = 0.008), strong and dull (*p* = 0.001), pale (*p* < 0.001), light grayish (*p* = 0.006), deep and light (*p* < 0.001), soft (*p* < 0.001), dull (*p* < 0.001), pale (*p* < 0.001), light grayish (*p* < 0.001), grayish (*p* = 0.009), light and dark (*p* = 0.001), soft and dark (*p* < 0.001), dull and dark (*p* < 0.001), pale and grayish (*p* = 0.002), and light grayish and grayish (*p* = 0.013). For the background color yellow-green, dark grayish was the highest, significantly higher than all the other tones (vivid, *p* < 0.001; bright, *p* < 0.001; strong, *p* < 0.001; deep, *p* < 0.001; light, *p* < 0.001; soft, *p* < 0.001; dull, *p* < 0.001; dark, *p* = 0.013; pale, *p* < 0.001; light grayish, *p* < 0.001; grayish, *p* < 0.001). Other combinations that showed significant differences were between vivid and dark (*p* = 0.004), bright and dark (*p* = 0.018), soft and dark (*p* = 0.029), light and dark (*p* < 0.001), grayish (*p* = 0.036), soft and dark (*p* = 0.001), dull and dark (*p* = 0.015), dark and pale (*p* < 0.001), light grayish (*p* < 0.001), and light grayish and grayish (*p* = 0.018).

## 4. Discussion

### 4.1. Examining the Effect of Hue by Saturation Range

First, the 12 tones were divided into three saturation ranges, and the effect of hue was compared for each saturation range. In the high-saturation range (Figure 1), which consists of vivid, bright, strong, and deep colors, the background color red increased the impression of anger more than yellow-green and blue-green. As expected, the higher the saturation level, the greater the hue effect. This is consistent with previous research conducted by [14] ([14]) and [22] ([22]). According to [2] ([2]), red has a direct effect on anger perception and identification because it is the color of the blood under our skin that becomes visible when we are angry. Similarly, [12] ([12]) examined whether facial expression recognition is affected by complexion, finding that a flushed face enhances anger recognition. In a study investigating emotional responses to color by presenting primary, intermediate, and achromatic colors selected from the Munsell color system ([8]), negative associations such as battle, blood, devil, and evil were evoked. Moreover, when we see a red face in a context that conveys a threat, our anger identification increases above baseline ([1]). In this study, the combination of an illustration of a red-faced angry expression and a red background color in the high-saturation range increased the participants’ feelings of anger. This is considered an expression of the impression of excitement and anger that red has as a hue effect and aligns with the findings of several previous studies. According to [8] ([8]), in the high-saturation range, blue-green alone evokes feelings of relaxation and security, while green evokes mainly positive emotional responses such as relaxation, composure, happiness, peace, and hope. Additionally, green light has a relaxing effect by making the parasympathetic nervous system dominant ([11]). Furthermore, yellow-green, which is similar in color, weakens the impression of anger. Thus, in the high-saturation range, differences in hue have the expected effect on anger.

However, in the low-saturation range, the colors’ vividness fades, and they become closer to gray, which highlights that the effect of differences in hue on anger was minimal. The fact that the effect of hue diminishes in the low-saturation range was also expected empirically, but the fact that a clear comparison was made for each saturation range, as in this study, and that the effect of hue is particularly low in the low-saturation range is an important discovery.

In the medium-saturation range (Figure 1), we observed different trends depending on the hue and value. For example, the combination with a soft red background, which had a higher value than a dull red background, showed a lower value. This may be because the lack of contrast between the background color and the face makes it difficult to see the face’s outline. Additionally, for the dull tone, blue-green enhanced anger more than any other tone, whereas red reduced anger more than any other tone. Thus, red and blue-green, which have exactly the same value for lightness and saturation and are located at opposite ends of the hue spectrum, have opposite effects on anger. This may be partly due to the complementary relationship between color and complexion. Moreover, the value of the complexion expressed in vivid red was the same as that of the background colors red and blue-green ([7]), which might have also affected the color-contrast relationship. However, why the medium-saturation range showed a different trend from those of the other saturation ranges is unclear. Although the factors involved are not easy to identify, the medium-saturation range has some effect on hue, and complex factors may be involved that cannot be explained by value, saturation, and hue alone.

### 4.2. Overall Saturation Effect

When the effect of the value on each saturation range was compared, low-value colors showed the strongest anger in all the saturation ranges. In particular, in the low-saturation range, all the colors lost their vividness and approached gray, such that the light tones resembled colors close to white, and the dark tones resembled colors close to black. Therefore, in the low-saturation range, the comparison was similar to that between shades of gray, and a low value of brightness was shown to increase anger. Furthermore, for dark grayish, which is the darkest hue in the low-saturation range, the value of red was slightly higher than that of blue-green. However, no differences were observed between it and yellow-green, and in the tone comparison, no differences were observed between the vivid red and dark grayish-red background colors. Furthermore, no significant differences were found between the vivid, deep, dark, and dark grayish-red backgrounds. This result could not have been predicted in relation to Question 2, and in the context of comparison, the low value of the color, as well as the high saturation of the red background, led to increased anger. In the high-saturation range, among the three tones of bright, soft, and deep, the background color with the lowest brightness, deep, evoked the strongest anger emotion. This result confirms the effect of brightness even in the high-saturation range and shows that low brightness enhances anger emotion.

The analysis of the red background in this high-saturation range produced some important results that should be considered in future research. The two tones, vivid and strong, both of which have a highly similar color to complexion, showed different values. Statistical significance was found between the two tones, which do not differ greatly visually, and vivid, which is completely assimilated with complexion, showed a higher value than strong. Whether the effect is due to the integration of complexion and background color, or the effect of the background color alone is difficult to discern from the results of this experiment. However, since no significant difference exists in the values of vivid and strong for the two colors other than the red background, denying the possibility that the integration of complexion and background color had an effect is impossible.

Furthermore, no significant difference was found in the value of deep for the three hues of the vivid background. This result suggests that vivid and deep in the high-saturation range indicate anger emotions that are particularly close to each other, and analyzing the combined effects of color saturation and brightness on emotions is necessary.

Brightness triggers a psychological response as a visual stimulus. [20] ([20]) and [13] ([13]) suggested that lower brightness is more likely to be associated with negative emotions. Additionally, visual information is limited in dark environments, and the ability to accurately identify potential threats is reduced, which can increase anxiety and stress, thereby leading to anger ([15]). [4] ([4]) conducted a survey of 20 men and 20 women to investigate their emotional responses to 10 main colors (white, pink, red, yellow, blue, purple, green, brown, black, and gray) and found that bright colors are associated with positive emotions, and dark colors are associated with negative emotions. However, white, pink, red, yellow, blue, purple, and green are considered bright colors, and brown, black, and gray are considered dark colors, and this is not brightness using the value as in this study. [21] ([21]). examined emotional recognition and intensity by adjusting the brightness of a light shining on a character modeled on a computer, finding that the brighter the character was lit, the more happiness it evoked, which made the subject appear more attractive and appealing.

Personal experiences also have a significant effect on individuals’ emotional responses to darkness. For example, people who had frightening experiences in the dark as children may continue to feel strong anxiety or fear of the dark as adults. Conversely, people who had positive experiences of feeling safe and protected in dark places as children may perceive the dark as relaxing. The living environment is also a potential influencing factor. People who live in urban areas are accustomed to bright environments even at night and are more likely to feel anxious in the dark. Conversely, people who live in rural areas with plenty of natural settings may be less resistant to the dark as darkness at night is a part of their daily lives. People who have grown up in natural environments and are familiar with the dark often perceive it as relaxing ([16]).

### 4.3. Color in Culture

If colors in the high-saturation range express the hue effect more strongly, then dealing with the medium-saturation range is complex and somewhat difficult. However, this complexity may have the potential to convey a rich message. The reason for the complexity observed in this study’s results may be that the participants were all Japanese. We provide some examples below of how the sense of color in the medium- and low-saturation ranges has developed in Japanese culture.

Commerce flourished in Japan during the late Edo period. The burgeoning merchant class enjoyed wearing colorful kimonos; however, the shogunate (the government of Japan at the time) issued a decree forbidding common people from being ostentatious or extravagant and wearing colors other than brown, gray, and indigo. People created many variations by being creative with sober shades of brown, black, and gray, and enjoyed these as stylish colors (high-value colors). This was called “48 browns and 100 grays” (meaning that many colors existed, not that the browns and gray were exactly 48 and 100). [5] ([5]) stated that the spirit of wabi-sabi was nurtured by Zen thought that arose within the culture of the samurai class during the Muromachi period and that the restrained color of “ke (everyday life)”, known as brown, became the epitome of aesthetic sense. Furthermore, a culture that appreciates a wide variety of subtle nuances of color was nurtured from dull colors with low saturation, limited by the prohibition of luxury.

The relationship between the cultural background and the value of light is an important factor in determining the effects of light and dark environments on individuals. In particular, emotional responses to a dark environment will likely diverge depending on each individual’s cultural background and personal experiences. Darkness can be perceived in two ways. Darkness is frequently depicted in horror films and literary works as an environment that symbolizes threat and danger, and it tends to be associated with fear and anxiety. However, darkness is also recognized as a means of promoting introspection and spiritual growth.

When the Japanese perception of darkness is considered, a complex and multifaceted aesthetic sense of dim light can be seen. Shadows are considered important aesthetic elements in traditional Japanese architecture and garden design. Tanizaki Junichiro discusses this in detail in his book, *In Praise of Shadows* ([17]). In Japan, which became increasingly Westernized, Tanizaki expressed the Japanese aesthetic of being lost through the soft light that shone through dark shadows. He analyzed in detail how the interplay between light and shadow is a core element of Japanese aesthetics in various cultural aspects, such as architecture, paper, food, makeup, and color, and highlighted its unique aesthetic value. Furthermore, in Japanese culture, dark environments are sometimes valued as meditative spaces that bring tranquility and peace. Traditional ceremonies, such as meditation and tea ceremonies, are still practiced today as a means of promoting spiritual peace ([9]); this unique sense of color for medium- and low-saturation ranges might have been passed down along with these traditions.

#### Limitations and Future Directions

As this study was conducted on Japanese people, comparing the data with those from regions or countries with different cultural backgrounds was not possible. Emotional responses to colors may differ depending on cultural factors, and generalizing the results of this study has certain limitations. For this reason, conducting international comparative studies that take cultural factors into account in future research is important. 

Additionally, in this study, the participants repeatedly evaluated the same illustrations with the same facial expressions, and this might have caused a shape-memory effect. This effect might have made the accurate evaluation of background color on emotion difficult. To overcome this issue, conducting experiments using multiple different illustrations and improving the research design by devising a more effective evaluation sequence and method will be necessary.

## 5. Conclusions

As communication methods change with the rapid progress of information and communication technology, effectively using color as a means of conveying visual information is becoming increasingly important. Although the development of artificial intelligence has made it easier to generate images and videos, humans must have knowledge of colors to use them appropriately. This study found that color value has a significant effect on anger; however, several questions regarding the relationship between color and emotion remain unanswered. Previous research has often focused on hue; however, in the future, more in-depth research on the direct visual impact of value and saturation will be necessary. Additionally, the phenomenon of a low value of lightness increasing anger is considered limited to anger, and different values would be shown for other emotions, such as joy and sadness. As [1] ([1]) noted, anger identification increases above baseline when faces are viewed in a threatening context, and the emotions that can be perceived from brightness are highly dependent on context. In future research, we would like to deepen our understanding of the relationship between visual information processing and emotion by conducting a detailed analysis of color and emotional responses and elucidating the complex factors in the medium-saturation range. This will allow us to explore new possibilities for emotional control through vision.

## Figures and Tables

**Figure 1 behavsci-15-00551-f001:**
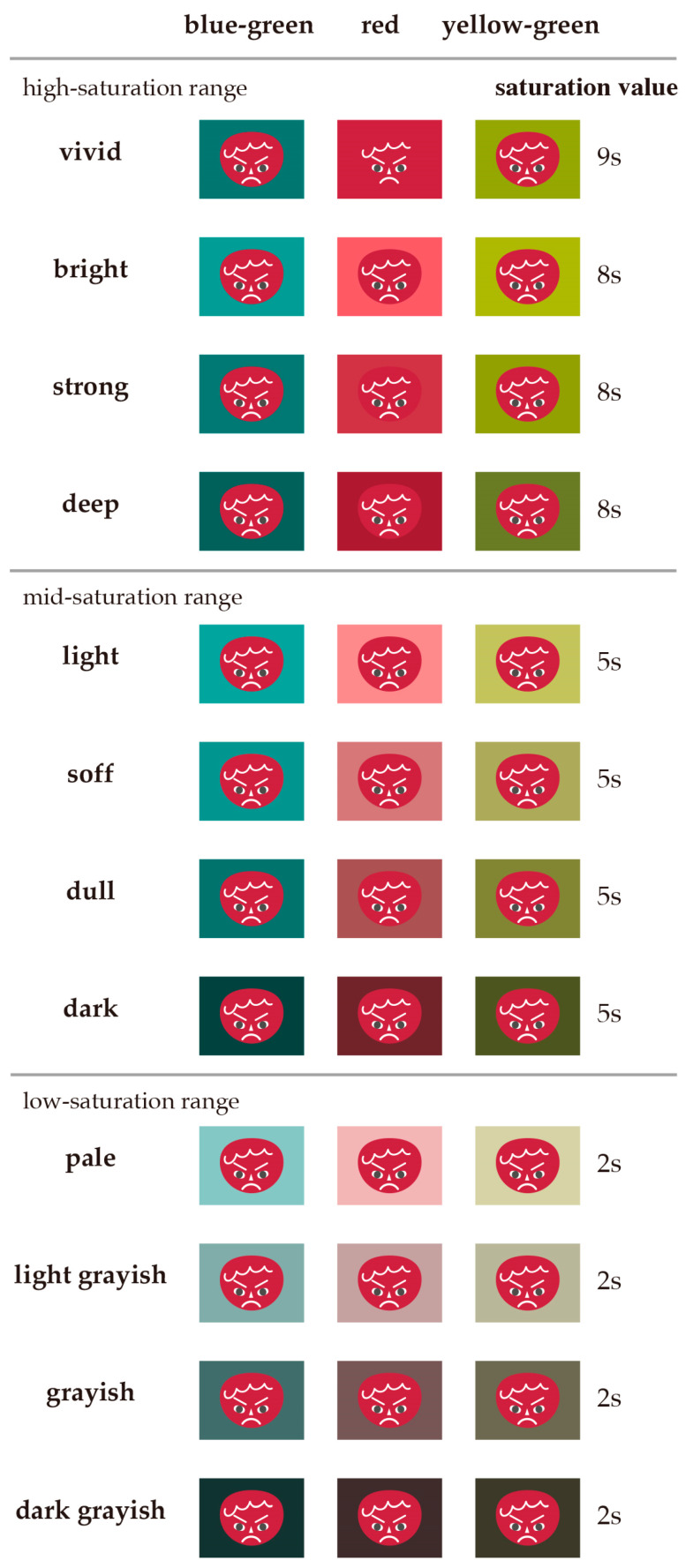
List of three background colors: red (2), yellow-green (10), and blue-green (14) × 12 tones, and red face (v2) combination. PCCS makes use of the characteristics of PCCS, where the saturation values are arranged in each saturation range. Vivid is 9s, high-saturation range 8s, medium-saturation range 5s, and low-saturation range 2s.

**Figure 2 behavsci-15-00551-f002:**
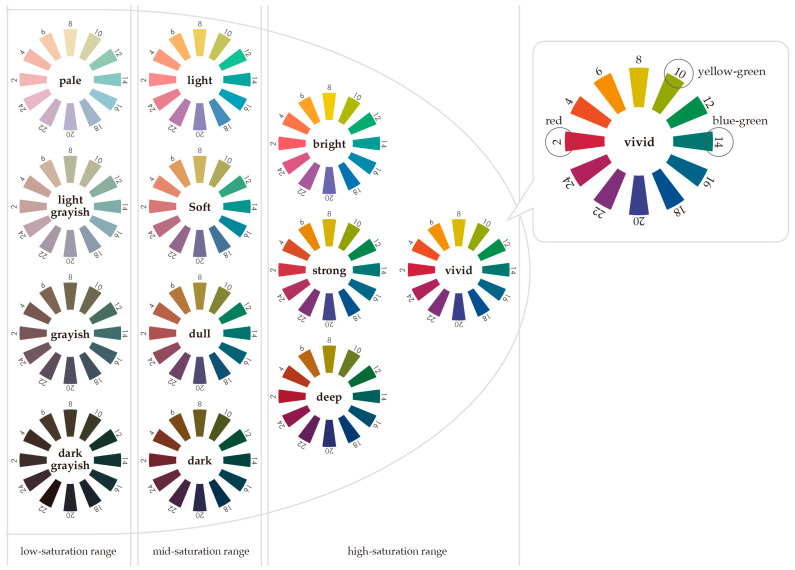
A 12-tone classification chart based on the PCCS color system. The tones are classified into high-, medium-, and low-saturation ranges. The illustration of the angry face is colored with a vivid tone of red (v2). The hue circle for vivid tones is shown in the upper right of the diagram, and the positions of red (2), yellow-green (10), and blue-green (14) are shown as background colors. The background colors were selected from the 36 colors (3 hues × 12 tones) of the PCCS color system.

**Figure 3 behavsci-15-00551-f003:**
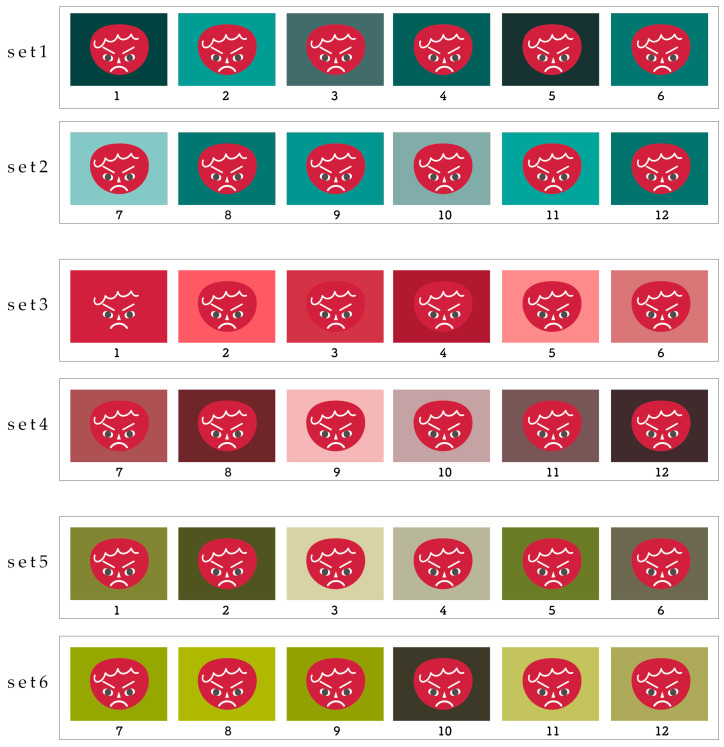
How the images were displayed during the survey: In the survey using Google Forms, six images were displayed at a time, from the top down, and the “anger” felt from the combination of the red angry face illustration and background color was rated on a scale of 1–6 or 7–12 for each image. The placement of the background color tones was not fixed but randomized for each hue to prevent the respondents from being biased in their answers by detecting a specific pattern or intention.

**Figure 4 behavsci-15-00551-f004:**
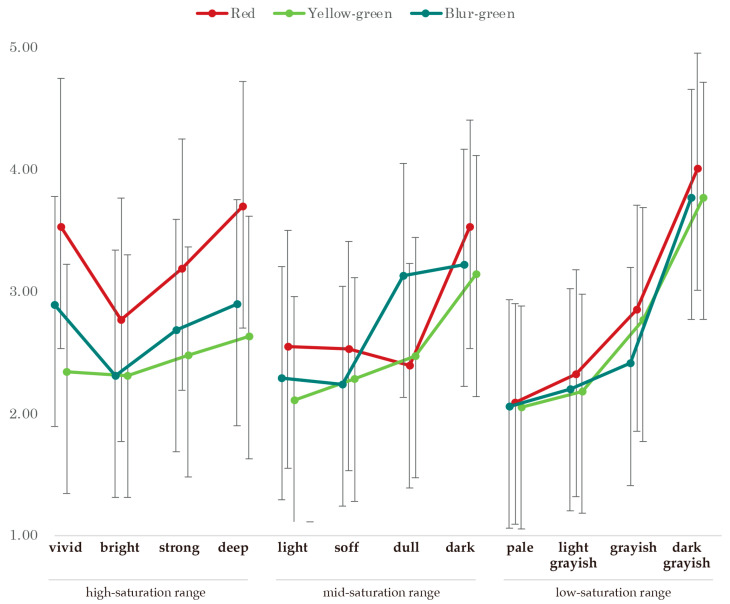
Graph of the effect of background color on angry faces. It shows that the lower the color brightness, the stronger the feeling of anger across all saturation ranges.

**Table 1 behavsci-15-00551-t001:** PCCS tone symbol–HSV value conversion chart.

Tone	Blue-Green		Red		Yellow-Green	
Vivid	v14	175°, 100, 49	v2	352°, 78, 77	v10	71°, 80, 71
Bright	b14	175°, 100, 63	b2	358°, 60, 87	b10	69°, 77, 76
Strong	s14	176°, 100, 45	s2	355°, 71, 72	s10	71°, 74, 64
Deep	dp14	179°, 100, 40	dp2	349°, 81, 60	dp10	74°, 69, 54
Light	lt14	171°, 44, 77	lt2	1°, 34, 93	lt10	68°, 42, 85
Soft	sf14	173°, 61, 61	sf2	0°, 39, 75	sf10	70°, 46, 68
Dull	d14	177°, 82, 44	s2	359°, 49, 59	d10	73°, 52, 52
Dark	dk14	180°, 95, 23	sk2	349°, 65, 35	dk10	80°, 53, 32
Pale	p14	168°, 17, 86	p2	358°, 7, 88	p10	68°, 17, 90
Light Grayish	ltg14	174°, 21, 68	ltg2	356°, 11, 71	ltg10	69°, 19, 73
Grayish	g14	180°, 29, 36	g2	354°, 24, 40	g10	82°, 21, 40
Dark Grayish	dkg14	180°, 25, 20	dkg2	349°, 27, 24	dkg10	74°, 20, 25

The correspondence between the PCCS tone symbols for each color (blue-green, red, and yellow-green) and the HSV values. From the left: tone name, blue-green tone number, blue-green HSV value, red tone number, red HSV value, yellow-green tone number, and yellow-green HSV value.

**Table 2 behavsci-15-00551-t002:** Table comparing differences by gender.

	Blue-Green			Red			Yellow-Green		
	Male	Female	N/A	Male	Female	N/A	Male	Female	N/A
v	3.00 ± 0.94	3.17 ± 0.88	2.50 ± 0.58	3.31 ± 1.26	3.77 ± 1.22	3.50 ± 0.58	2.31 ± 0.97	2.73 ± 0.86	2.00 ± 0.00
b	2.42 ± 1.03	2.75 ± 1.02	1.75 ± 0.50	2.85 ± 1.16	2.97 ± 0.96	2.50 ± 0.58	2.42 ± 1.07	2.49 ± 0.99	2.00 ± 0.00
s	2.73 ± 1.00	3.07 ± 0.87	2.25 ± 0.50	3.12 ± 1.18	3.21 ± 1.05	3.25 ± 0.50	2.46 ± 0.91	2.73 ± 0.89	2.25 ± 0.50
dp	2.96 ± 0.96	2.99 ± 0.84	2.75 ± 0.50	3.54 ± 1.07	3.81 ± 1.03	3.75 ± 0.50	2.69 ± 0.97	2.95 ± 0.99	2.25 ± 0.50
lt	2.35 ± 0.98	2.54 ± 0.91	2.00 ± 0.00	2.73 ± 1.08	2.68 ± 0.94	2.25 ± 0.50	2.27 ± 0.96	2.31 ± 0.83	1.75 ± 0.50
sf	2.27 ± 0.96	2.44 ± 0.77	2.00 ± 0.00	2.62 ± 1.02	2.72 ± 0.86	2.25 ± 0.50	2.38 ± 0.85	2.44 ± 0.85	2.00 ± 0.00
d	3.00 ± 0.98	3.38 ± 0.91	3.00 ± 0.00	2.46 ± 0.95	2.70 ± 0.81	2.00 ± 0.00	2.69 ± 1.16	2.71 ± 0.94	2.00 ± 0.00
dk	3.19 ± 1.10	3.46 ± 0.92	3.00 ± 0.00	3.38 ± 1.06	3.70 ± 0.83	3.50 ± 0.58	3.04 ± 1.08	3.38 ± 0.96	3.00 ± 0.00
p	2.12 ± 1.03	2.07 ± 0.85	2.00 ± 0.00	2.31 ± 1.01	2.22 ± 0.76	1.75 ± 0.50	2.15 ± 0.88	2.23 ± 0.83	1.75 ± 0.50
ltg	2.08 ± 1.02	2.27 ± 0.78	2.25 ± 0.50	2.31 ± 1.01	2.39 ± 0.83	2.25 ± 0.50	2.23 ± 0.91	2.30 ± 0.79	2.00 ± 0.00
g	2.54 ± 0.91	2.70 ± 0.76	2.00 ± 0.00	2.69 ± 0.93	2.86 ± 0.86	3.00 ± 0.00	2.81 ± 1.06	3.01 ± 0.89	2.50 ± 0.58
dkg	3.69 ± 1.09	3.87 ± 0.84	3.75 ± 0.50	3.73 ± 1.19	4.05 ± 0.89	4.25 ± 0.50	3.62 ± 1.10	3.94 ± 0.91	3.75 ± 0.50

Note: Abbreviations are as follows: v = vivid; b = bright; s = strong; dp = deep; lt = light; sf = soft; d = dull; dk = dark; p = pale; ltg = light grayish; g = grayish; dkg = dark grayish.

**Table 3 behavsci-15-00551-t003:** Table of background color effects for angry faces.

	Blue-Green		Red		Yellow-Green	
	Mean	SD	Mean	SD	Mean	SD
high-saturation range						
Vivid	2.89	0.16	3.53	0.22	2.34	0.16
Bright	2.31	0.18	2.77	0.18	2.31	0.18
Strong	2.68	0.16	3.19	0.19	2.48	0.16
Deep	2.90	0.16	3.70	0.19	2.63	0.18
medium-saturation range						
Light	2.29	0.17	2.55	0.17	2.11	0.15
Soft	2.24	0.15	2.53	0.16	2.28	0.15
Dull	3.13	0.17	2.39	0.15	2.47	0.18
Dark	3.22	0.17	3.53	0.16	3.14	0.18
low-saturation range						
Pale	2.06	0.16	2.09	0.15	2.05	0.15
Light grayish	2.20	0.15	2.32	0.16	2.18	0.15
Grayish	2.41	0.14	2.85	0.16	2.77	0.17
Dark grayish	3.77	0.16	4.01	0.17	3.77	0.17

## Data Availability

The datasets used and/or analyzed during the current study are available from the corresponding author upon reasonable request.

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
