# Peer review of "Effects of Saturation on Anger in a Low-Saturation Range: A Comparison of Background Colors in 12 Tones"

_behavsci, 2025, doi:10.3390/bs15040551_

Round 1

Reviewer 1 Report (Previous Reviewer 1)

Comments and Suggestions for Authors

The article explores the relationship between colour saturation and emotional perception. 
Progress has been made in improving the inclusion of literature to establish an appropriate theoretical framework. However, studies of some antiquity have been used, which is an improvement to be made.

I find an important shortcoming in the analysis of the results, as no analysis is established taking into account demographic, cultural or gender perspective data. For example, are there significant differences between men and women? This type of analysis is essential to establish rigorous research.

I therefore propose a major revision of the work presented.

Author Response

Date: 28th Mar 2025

To:  Reviewer #1,

From:  Akinori Shimodaira

Re.: 

Title:  Effects of Saturation on Anger in a Low-saturation Range: A Comparison of Background Colors in 12 Tones

Thank you for your revision invitation letter with reviewer’s comments.

Point-by-point replies to the comments are presented in this letter.

The parts in red in the manuscript of the paper are the parts that have been added or corrected. In this response, we have written supplementary comments on the corrected parts.

We hope this revision will be satisfactory for the Reviewers as well as your editorial team, and our paper will be accepted for publication in the Behavioral Sciences.

We look forward to hearing from you at your earliest convenience.

Sincerely yours,

Akinori Shimodaira

Graduate School of Science and Technology Doctoral Program of Advanced Fibro-Technology, Kyoto Institute of Technology, 1 Hashikami-cho, Matsugasaki, Sakyo-ku, Kyoto 606-8585, Japan;

Email:  as_st@murgraph.com, d8851002@edu.kit.ac.jp

Reply to the comments from Reviewer #1

  1. The article explores the relationship between colour saturation and emotional perception.

Progress has been made in improving the inclusion of literature to establish an appropriate theoretical framework. However, studies of some antiquity have been used, which is an improvement to be made.

I find an important shortcoming in the analysis of the results, as no analysis is established taking into account demographic, cultural or gender perspective data. For example, are there significant differences between men and women? This type of analysis is essential to establish rigorous research.

I therefore propose a major revision of the work presented.

<Reply>

<1>

Response regarding improvements to the recorded documents

Thank you for pointing this out. In response to your comment, Wexner (1954) has been changed to Valdez, P. and Mehrabian, A. (1994).

(Page1, Lines 36-41)

(Page14, Lines 471-472)

<2>

Response to the comment about gender differences

Thank you for pointing this out. In the data for this study, we examined gender differences among the 26 males and 99 females, excluding the 4 who did not respond, and found no gender differences. Therefore, we conducted a variance analysis without gender differences as a factor. We have added this to the results section.

(Page7, Lines 208-210)

Reviewer 2 Report (Previous Reviewer 2)

Comments and Suggestions for Authors

Dear Author,

No further comments. Thank you for addressing my previous concerns/comments.

Best wishes.

Author Response

Date: 28th Mar 2025

To:  Reviewer #2,

From:  Akinori Shimodaira

Re.: 

Title:  Effects of Saturation on Anger in a Low-saturation Range: A Comparison of Background Colors in 12 Tones

Thank you for your time and effort in reviewing my manuscript. I sincerely appreciate your insightful comments and suggestions, which have been invaluable in improving the quality of the paper.

Best regards,

Akinori Shimodaira

Graduate School of Science and Technology Doctoral Program of Advanced Fibro-Technology, Kyoto Institute of Technology, 1 Hashikami-cho, Matsugasaki, Sakyo-ku, Kyoto 606-8585, Japan;

Email:  as_st@murgraph.com, d8851002@edu.kit.ac.jp

Reviewer 3 Report (Previous Reviewer 3)

Comments and Suggestions for Authors

The authors have addressed my concerns in the revised manuscript. The manuscript has

been sufficiently improved and now warrants publication. I have no other comments.

Author Response

Date: 28th Mar 2025

To:  Reviewer #3,

From:  Akinori Shimodaira

Re.: 

Title:  Effects of Saturation on Anger in a Low-saturation Range: A Comparison of Background Colors in 12 Tones

Thank you for your time and effort in reviewing my manuscript. I sincerely appreciate your insightful comments and suggestions, which have been invaluable in improving the quality of the paper.

Best regards,

Akinori Shimodaira

Graduate School of Science and Technology Doctoral Program of Advanced Fibro-Technology, Kyoto Institute of Technology, 1 Hashikami-cho, Matsugasaki, Sakyo-ku, Kyoto 606-8585, Japan;

Email:  as_st@murgraph.com, d8851002@edu.kit.ac.jp

Round 2

Reviewer 1 Report (Previous Reviewer 1)

Comments and Suggestions for Authors

To accept the statement in the results: ‘When we examined gender differences based on the data from this study, we found 207 no gender differences, so we conducted a variance analysis without gender differences as 208 a factor’, some table or graph showing this statement should be included. In addition, demographic data should be included in the study.

Author Response

Date: 10th Apr 2025 To: Reviewer #1, From: Akinori Shimodaira Re.: Title: Effects of Saturation on Anger in a Low-saturation Range: A Comparison of Background Colors in 12 Tones Thank you for your revision invitation letter with reviewer’s comments. Point-by-point replies to the comments are presented in this letter. The parts in red in the manuscript of the paper are the parts that have been added or corrected. In this response, we have written supplementary comments on the corrected parts. We hope this revision will be satisfactory for the Reviewers as well as your editorial team, and our paper will be accepted for publication in the Behavioral Sciences. We look forward to hearing from you at your earliest convenience. Sincerely yours, Akinori Shimodaira Graduate School of Science and Technology Doctoral Program of Advanced Fibro-Technology, Kyoto Institute of Technology, 1 Hashikami-cho, Matsugasaki, Sakyo-ku, Kyoto 606-8585, Japan; Email: as_st@murgraph.com, d8851002@edu.kit.ac.jp  Reply to the comments from Reviewer #1 To accept the statement in the results: ‘When we examined gender differences based on the data from this study, we found no gender differences, so we conducted a variance analysis without gender differences as a factor’, some table or graph showing this statement should be included. In addition, demographic data should be included in the study. Thank you for pointing this out. In order to describe the differences between the genders, a new three-factor analysis of variance was conducted. The results of this were added, and a table showing the mean values and standard deviations for each gender for each combination of hue and tone was added. (Page6 Lines 160-161) (Page8, Lines 208- Page10, Lines 210)

Round 3

Reviewer 1 Report (Previous Reviewer 1)

Comments and Suggestions for Authors

The gender perspective has been integrated, analysed and shows the results and conclusions.

This manuscript is a resubmission of an earlier submission. The following is a list of the peer review reports and author responses from that submission.

Round 1

Reviewer 1 Report

Comments and Suggestions for Authors

The work presents an important deficit in the analysis of the existing literature on the problems it raises. Therefore, it does not stand up in the state of the art, nor in the discussion and conclusions.

In the methodology and subsequent analysis of the results, the analysis does not consider the demographic data of the sample, which makes it an incomplete work.

I recommend the authors to improve the literature review to support the statements and build a rich discussion, to use the demographic data of the sample and to enrich the work with more graphs.

Comments on the Quality of English Language

some typos have been found

Author Response

Please kindly check the attachment

Reviewer 2 Report

Comments and Suggestions for Authors

Apologies to the authors and the editorial team for the delay in my review. I thank you for your patience.

I commend the authors on this novel study investigating the effects of background hue, saturation, and value on rating a pictorial angry face. Previous research has typically focused on the effects of hue on angry face ratings whilst controlling saturation and value. The study found that lower brightness resulted in increased anger ratings, in addition low saturation resulted in similar anger ratings regardless of the hue.

Please see below more specific comments.

Abstract

- It is important to specify that the faces are not images of human facial expressions but rather a pictorial angry face.

Introduction

- The introduction would benefit from definitions of hue, value, and saturation.

- Figures 1 and 2 should be made clear – they are currently blurry.

- Also, the specification of the HSV values of the background colours is vital for this study.

- A point to consider, although the value and saturation may be the same across hues, the perceptual brightness of the backgrounds may be different.

- The justification behind the choice of hues (yellow-green; blue-green) is unclear.

- I believe the following paper may be of value to further assist authors in the development of their hypotheses and for their discussion.
Wilms, L., & Oberfeld, D. (2018). Color and emotion: effects of hue, saturation, and brightness. Psychological research82(5), 896–914. https://doi.org/10.1007/s00426-017-0880-8

Method

Section 2.1

- Was colour-blindness data obtained from these participants?

- Line 123: “the survey was conducted in a way that depended on the display environment of the participants.” It is unclear what is meant by this, more information would be appreciated.

Section 2.2

- It would be useful to make the preliminary survey results available on OSF. Also it is unclear what information this survey collected.

- The numbers next to each colour should be linked to the figure more clearly.

- Line 133: It is unclear how participants were informed which background to rate if they were all presented simultaneously.

- The same pictorial angry face is repeated on different backgrounds, how might this repetition influence the responses? What is the difference between this design and asking participants to rate colours as evoking more or less anger? This needs to be addressed in the discussion.

- How were the rating scales coded?

- How long did the entire study take to complete?

- Why not include a picture of a actor posing an angry facial expression? That may be more ecologically valid.

- It seems that the angry face blends in with the red background (e.g., vivid red), making the entire shape look angry and thus potentially exaggerating the anger rating. This might not be comparable with the other blue-green and yellow-green backgrounds.

- Do you think having a black border around the face would be a way to counter the blending in of the face with the background? - Or alternatively, having the background colour on the angry face such that the pictorial facial expression is present without the redness of the face across all conditions. This would still be interesting to investigate and may better control for any contrast effects.

Results

- Type of analysis is appropriate.

- Line 147: check APA formatting

- Line 163: p-values missing for light grayish.

- The results section is very text-heavy, I wonder whether a graph might more easily depict the differences in ratings.

Discussion

- Table 1: Include a note underneath the table indicating what the numbers represent.

- I disagree with statement made on line 206, the authors cannot conclude that participants were unable to perceive the differences in the hue of the background based of the ratings provided for the angry faces. The current methodology does not provide information about whether or not participants were able to perceptually differentiate between the background colours.

- Line 249: Citation needed.

- The discussion section might benefit from a more focused discussion of the findings. It seems in its current state to be drawing away from the findings to physiological aspects of colour vision which, although related to the study, is not directly relevant.

- Further, there is expansive literature on the top-down effects on feeling – line 276 onwards. This does not seem to be very relevant in the discussion.

- The section on colour in culture is interesting. It may then be important to conduct further investigation into individual differences in the associations of bright and dark to further understand the complexities of the findings.

- Future research has been clearly specified. I would like to see clearer discussions of the strengths and limitations of this study.

Inclusion of key methodological details and a more careful organisation of ideas and interpretation in the discussion section is recommended.

Comments on the Quality of English Language

Line 10: should be ‘face’ not ‘faces’.

Line 273: Typo

Author Response

Please kindly check the attachment

Reviewer 3 Report

Comments and Suggestions for Authors

Using an online survey, the current study investigated how background hues affected the perception of angry red faces under different saturation conditions. The main results demonstrated that, although the difference effects of hue on anger were found in the high-saturation range, the effects failed to be observed in the low-saturation range. The design of the method and analyses are appropriate, and the manuscript is well written. This study provided interesting data, and extended our insights about how color affects emotion processing. Several comments were outlined below.

Comments:

Line 32. “The strong influence of color on emotions is widely recognized.” It would be good to add associated literatures. And, “Few studies have examined these elements in combination.”, associated literatures should be also added. In other some places associated literatures were also absent.

Line 110, 114. Texts and pictures in the two figures (Figure 1 and and Figure 2) became blurry from my computer. Please double-check them.

Line 144. To visually observe how differences in value within the same hue affected anger ratings, it would be good to draw line plots when the results were presented.

Line 132, How the 36 images were arranged when they were presented to the participants? Were different arrangements made across participants? It could be good to describe them in more details.

Line 147, “F 22, 2816 = 18.786, p < .001” should be “F (22, 2816) = 18.786, p < .001”. The latter is more standard in the use of statistical symbol.

Line 326. It would be good to add a paragraph describing the limits and future work, rather than in the Conclusion section. Brief descriptions are appropriate for the Conclusion section.

Author Response

Please kindly check the attachment
